# Physical Unclonable Functions in the Internet of Things: State of the Art and Open Challenges

**DOI:** 10.3390/s19143208

**Published:** 2019-07-21

**Authors:** Armin Babaei, Gregor Schiele

**Affiliations:** Embedded Systems Group, Faculty of Engineering, University of Duisburg-Essen, 47048 Duisburg, Germany

**Keywords:** PUF, IoT, FPGA, security, hardware security

## Abstract

Attacks on Internet of Things (IoT) devices are on the rise. Physical Unclonable Functions (PUFs) are proposed as a robust and lightweight solution to secure IoT devices. The main advantage of a PUF compared to the current classical cryptographic solutions is its compatibility with IoT devices with limited computational resources. In this paper, we investigate the maturity of this technology and the challenges toward PUF utilization in IoT that still need to be addressed.

## 1. Introduction

Security is one of the main current challenges for the Internet of Things (IoT) [1,2,3] as it begins to materialize in our personal life and future industrial systems (Industry 4.0) [4]. IoT devices must face several tough challenges, such as low energy consumption, lack of computational resources [5,6], as well as the need to secure devices against cyber-attacks [7]. Unfortunately, energy footprint concerns, as well as the scarcity of computational resources restrict which cryptographic methods can be implemented on these devices. This makes it difficult to implement traditional security mechanisms like an asymmetric handshake, which requires hashes and asymmetric cryptography [8]. Early attacks show that vendors are often unaware of how easy it is to attack their products, e.g., allowing remote updates on an unprotected Telnet port [9]. At the same time, IoT devices must often be inexpensive, increasing the challenge to implement security functions for them.

Authentication, authorization, and privacy are three sides of the security triangle in IoT. Authentication is the first barrier in front of cyber-attacks. Physical Unclonable Functions (PUFs) have been proposed as a lightweight, cost-efficient, and ubiquitous solution. Importantly for IoT developers, PUFs promise to achieve perfectly secure authentication without any cryptographic assets on the device, which makes them especially interesting for resource-scarce IoT devices [10,11]. However, newer research indicates that PUFs may not be able to fulfil all initial promises, and more effort is needed to realize truly secure solutions with them.

In this paper, we provide an overview of the current state of the art of PUFs for IoT systems, specifically for authentication. Our main contribution is a critical discussion of how well current PUF architectures and PUF-based protocols fulfil the original promise, namely to realize security without cryptographic assets, i.e., with little resource overhead. We analyze specific security requirements for the IoT domain and how PUFs can be used to fulfil them. Our focus is on approaches that are well-established and understood regarding their advantages and disadvantages. We mention other approaches briefly if we feel that they have the potential to become important for the IoT in the future. A PUF can be realized in IoT products by integrating a special PUF circuit, e.g., as a stand-alone ASIC or as part of a system on chip. The other option is to implement a PUF circuit in a reconfigurable hardware like a Field Programmable Gate Array (FPGA). This second approach gives the IoT developers more flexibility about the PUF architecture used and allows them to tailor the system more closely to their specific application needs. Therefore, in this paper, we concentrate mainly on PUFs implemented on FPGAs and only briefly discuss PUFs implemented as PUFs.

The paper is structured as follows. After an initial discussion of PUF basics in Section 2, we analyze PUF-based threats on IoT devices (Section 3). Then, in Section 4, we discuss possible defense strategies and present existing PUF architectures (Section 5), as well as existing authentication protocols using PUFs (Section 6). In Section 7, we assess how far we are in fulfilling the promise of PUFs in the IoT and conclude the paper in Section 8.

## 2. PUF Initial Definitions

Like humans, each chip has its own fingerprint, which is created during manufacturing. This intrinsic characteristic is extractable by adding a specific circuit architecture, a so-called PUF circuit, to the chip (see Figure 1). PUF circuits receive a sequence of bits (alleged challenges) as the input and generate a sequence of bits (so-called responses) as the output. No two chips generate identical responses for a particular challenge. The combination of a challenge and its corresponding response is called a Challenge Response Pair (CRP).

PUF circuits are implemented using different CMOS technologies, most prominently on a memory chip, as an Application-Specific Integrated Circuit (ASIC) or on an FPGA. There are two types of PUF circuits, weak and strong PUFs. For strong PUFs, increasing the size of the PUF circuit leads to an exponential growth in the number of CRPs. For weak PUFs, it will increase linearly [12,13].

PUFs generally are used for authentication and secure communication. Since PUF-based authentication does not require classical cryptographic assets, it fits properly into the resource demands of IoT devices. PUF authentication protocols have two phases: the enrollment and the authentication phase [14]. The first proposed PUF-based authentication protocol [15] works as follows: During the enrollment phase (see Figure 2a), the chip which contains the PUF circuit is directly connected to the server. The server sends challenges, and the PUF circuit sends back the responses. The server stores all CRPs in a table. Then, the chip will be mounted into the IoT device. During the authentication phase (see Figure 2b), if the device needs to be authenticated by the server, the server sends an arbitrary PUF challenge to the device. The device measures the PUF and sends back the generated response bits. If the measured response matches the stored response in the server database, then the device is authenticated. Another traditional application for a PUF is to extract a key from the PUF response to establish an encrypted communication [16,17]. This is out of the scope of this paper.

## 3. PUF-Based Threats on IoT Devices

In this section, we look into possible threats for IoT devices that use a PUF for authentication. IoT devices can potentially be installed in unprotected areas. As a result, they are exposed to a variety of threats, ranging from direct physical attacks (e.g., hammering) to disrupting communication [18] and manipulating the physical device operation conditions [19]. More traditional attacks like trying to read out secret keys from memory and communication attacks [20] are possible as well.

From the viewpoint of a PUF-based security system, the main new threat is for an attacker to gain the ability to provide the correct response for a given challenge. This can, e.g., be done by constructing a physical clone of the PUF [21] or by predicting CRPs with a modeling attack [13,22]. We discuss both of these approaches in more detail later. We consider two different attacker models. In the first one, the attacker is able to intercept the communication of devices. In the second one, an adversary has physical access to the device.

### 3.1. Man in the Middle Attack

In this type of attack, an adversary can overhear the communication channel between a server and a device and intercept and store exchanged CRPs. CRPs can then be used directly to perform replay attacks [11] or indirectly by feeding them into a machine learning algorithm and learning a model of the PUF that can predict other CRPs.

Man in the middle attacks can be performed with comparatively little effort in the IoT since devices often connect dynamically to previously unknown devices. An adversary can place an inexpensive computer like a Raspberry Pi in proximity to the attacked device and let it join the same (possibly encrypted) wireless network. Therefore, the risk for such an attack taking place is relatively high, and suitable defense mechanisms must be used.

### 3.2. Side Channel Attack

In this type of attack, the adversary has physical access to the device. Side channel attacks on PUF can be classified based on two orthogonal axes [23]. On the first axis, we can distinguish between invasive, semi-invasive, and non-invasive attacks. On the second axis, attacks can be either active or passive.

#### 3.2.1. Invasive, Semi-invasive, and Non-Invasive

Invasive attacks can be performed by damaging the chips and gaining direct access to the internal components. The initial assumption was that PUFs were immune against such attacks [24] because they would damage the structure of a PUF, rendering it in-operational. There is no stored on-chip secret key, which the attacker can extract [25]. However, there are reports that prove that PUFs are in fact vulnerable against invasive attacks [21,26], including creating a full physical clone of a PUF [27]. Invasive attacks are much more complicated and costly. An adversary usually has to move the attacked IoT device into a specialised lab, where expensive laboratory equipment is available. This makes this type of attack less attractive for IoT devices, especially when the devices are located in public places and bringing them to a laboratory is not possible.

For semi-invasive attacks, an adversary requires access to the chip surface; however, the attack will not damage the passivation layer of the chip [28]. Photonics emission [29] and electromagnetic probing [30,31] are reported semi-invasive PUF attacks. Although semi-invasive attacks use simpler techniques than invasive attacks, they still require special equipment in a lab.

Non-invasive attacks try to extract secret information by exploiting data (e.g., power consumption, delay time) without direct access to internal components. The equipment needed for these attacks is relatively small and inexpensive [32] and can even be transported and installed near the attacked IoT devices. Non-invasive attacks use machine learning algorithms as an analysis tool and can duplicate CRPs with a very high level of accuracy [33,34,35,36].

#### 3.2.2. Active and Passive

Active attacks actively tamper with the system, e.g., by modifying the supply voltage Vcc or the operational temperature [19] to perform attacks on the PUF. In contrast, passive attacks passively observe data, e.g., the temperature or energy consumption of a PUF to attack it. Both types of attack require physical access to the device and have been shown to be applicable to attacking of PUFs [32].

## 4. Defense Strategy

In the previous section, we characterized attacks on PUF-based IoT devices. Every day a new attack is proposed. Now, we discuss possible defense strategies for the different classes of attacks. We start with side channel attacks. Invasive and semi-invasive side channel attacks require direct access to the PUF. If the adversary can take the device to a lab, (s)he can use sophisticated machinery to attack the system. Adequate physical protection mechanisms for IoT devices (e.g., epoxy adhesive or gluing the PCB) and anomaly detection (e.g., by detecting unusual device mobility via inexpensive gyroscope sensors) can mitigate the threat level, but cannot provide perfect protection against invasive and semi-invasive side channel attacks. Note, however, that these types of attacks are usually expensive, and cost efficiency can be considered as a barrier for low-cost IoT systems. Non-invasive attacks, on the other hand, can work with comparatively simple equipment outside of a lab and without attacking the physical protection of the device. Passive non-invasive attacks apply a series of challenges to the PUF and monitor external factors like power consumption to create a machine learning-based model of the PUF. To do so, a large number of challenges is necessary, ranging in the thousands or even hundreds of thousands [32]. To counter this, the PUF can be constructed such that it either only accepts very specific challenges, making it very difficult to obtain enough of them, or to only accept a small number of challenges per second, slowing down the attack by several orders of magnitude [13,37].

Active non-invasive attacks additionally exploit the fact that PUFs might behave differently under different operational conditions. By manipulating these conditions they can, e.g., reduce the number of usable CRPs and make modeling attacks easier. Despite defending against the modeling attack itself, a suitable defense strategy is to make the PUF more robust against external conditions, essentially reducing active attacks to the same class as passive attacks. For the man in the middle attacks, avoiding the reuse of CRPs [11,38] is a well known solution for replay attacks. There are two options: first, CRPs can be kept secret by encrypting them [17]. This is especially suitable if the number of CRPs is very limited, e.g., for weak PUFs, but costs extra computational resources, mitigating our original reason to use PUFs in the IoT. Alternatively, we can choose a (strong) PUF design with an amount of CRPs that is large enough to never reuse one. The classic tactic to have a larger number of CRPs is to occupy more computational resources on the chip [12,19]. Since computational resources are scarce in IoT devices, this approach may not fit properly. Recently, an approach using a spatial reconfigurable PUF [7] has been introduced as a solution to increase the number of CRPs without increasing the size of the PUF circuit on the chip. Note that we can combine strong PUFs with encrypting CRPs for additional protection. However, this will again induce additional overhead.

In more advanced man in the middle attacks, an adversary again uses machine learning to perform a modeling attack and predict CRPs [22], this time using previously intercepted CRPs as input into the machine learning algorithm. Again, thousands of CRPs are needed to perform successful attacks [13]. One approach to address these attacks was to enhance the PUF architecture robustness by increasing non-linearity to the architecture [13,27,39]. Unfortunately, this strategy ultimately proved to be unsuccessful [22]. Another approach was to use cryptographic methods [16,40,41] to handle modeling attacks via machine learning. This approach had more success, but neglected the PUF architecture, opening the area for side channel attacks [19,35].

We argue that to develop a secure PUF-based IoT system, both side channel attacks and man in the middle attacks must be addressed simultaneously. As discussed before, physical protection should be used to counter invasive and semi-invasive side channel attacks. In addition, the most important non-invasive side channel attack for IoT devices is to modify environmental parameters, e.g., to increase the PUF error rate and, thus, reduce the number of useable CRPs. Higher robustness can be achieved by making the PUF architecture robust against such environmental variations, and the architecture of the used PUF shall mitigate the risk for non-invasive side channel attacks. Therefore, selecting an appropriate PUF architecture is crucial. Finally, to counter the man in the middle attacks, the PUF has to be combined with a suitable authentication protocol to realize an integrated approach.

In the following sections, we investigate existing PUF architectures and authentication protocols for IoT devices and analyze their robustness against attacks.

## 5. Physical Unclonable Function Architecture Selection for IoT

Selecting an appropriate PUF architecture for IoT applications is not trivial. The main criteria for this selection are:Robustness against possible attacks.Good statistical properties are important features in cryptographic applications (CRPs uniqueness and uniformity).Number of CRPs vs. occupied area (strong PUF: exponential increment in the number of CRPs by increasing utilized computational resources vs. weak PUF: linear increment in the number of CRPs by increasing utilized computational resources).Easy implementation process on FPGA (making it possible to adapt the PUF after deploying the IoT devices and updating it to new counter attacks).

In this section, we discuss the proposed PUF architectures for IoT applications, and we look into the strengths and weaknesses of each architecture. Figure 3 provides an overview.

### 5.1. Arbiter PUF

An Arbiter PUF [42] as shown in Figure 4 works by comparing two delay paths of equal length and generates a “0” or a “1” as the result of this comparison. Although the two paths should have the same delay, due to microscopic variations, one of them is actually faster than the other. The challenge is used to determine the two paths that are used dynamically. To do so, the individual bits of the challenge are used as the input into a series of interconnected multiplexers. Depending on the input, each multiplexer decides the next multiplexer to switch its output. This generates numerous combinations of possible paths.

Arbiter PUFs are strong PUFs (Criterion 3), and the authors of [38,43,44] proposed an arbiter PUF for IoT applications. They are quite sensitive to the delay paths, i.e., all delay paths should have precisely the same length to achieve good statistical properties (Criterion 2) [16,17]. Although it is possible to implement this type of PUF on an FPGA [38], the implementation of an Arbiter PUF is cumbersome [17] (Criterion 4) and better suited for ASICs. Furthermore, active and passive side channel attacks on the arbiter PUF are reported (Criterion 1) [32]. An XOR arbiter PUF is a modification to provide robustness against machine learning [45]; however, further experiments [22,46] showed that a stand-alone XOR arbiter PUF cannot have full resiliency against machine learning.

### 5.2. Ring Oscillator PUF

A Ring Oscillator (RO) PUF is built from several identical ring oscillators (see Figure 5). Suppose there are 2∗m ROs, and each one of them oscillates with its own frequency $f_{a}$ (for the first RO) to $f_{2}m$ (for the last RO). By comparing the two RO frequencies, either “0” or “1” will be generated. Each incoming challenge determines the pair of ROs that is used. In theory, all ROs’ oscillation frequencies should be the same; however, manufacturing process variations caused slight differences in this term. Note that RO PUFs that use such non-exclusive RO comparisons may lead to bias responses. A solution for this issue is discussed in [47].

RO PUFs are strong PUFs (Criterion 3), and the main reason behind using them for FPGA-based IoTs is their easy implementation on FPGA (Criterion 4). They have good statistical properties (Criterion 2) [7,17]. One of the drawbacks of RO PUFs is sensitivity to environmental conditions. As an example, a response bit flip can happen due to temperature variation if RO pairs have adjacent frequencies (see Figure 6). This makes RO PUFs a good candidate for active side channel attacks (Criterion 1). Although a solution for this has been proposed [48], it is only applicable to ASICs.

### 5.3. SRAM PUF

One of the most important PUFs in terms of statistical properties (Criterion 2) and reliability are SRAM PUFs [16,17,49] (see Figure 7). This architecture is commercially available [50] and is based on the power-up state of SRAM blocks. When an SRAM PUF powers up, the initial value of each single SRAM cell can be either “0” or “1”. This power-up state will be different from the state of any other device powering up [51,52,53]. Therefore, the challenge can be used as the address of an SRAM cell, and the initial value of it can be used as the SRAM PUF response. It is also possible to implement SRAM PUFs on micro-controllers [54,55,56], but implementing them on FPGAs is not straightforward (Criterion 4). This makes them an unpopular choice for FPGA-based PUFs for the IoT [7,17,57]. An SRAM PUF is a weak PUF (Criterion 3) with a limited number of CRPs. Therefore, it needs obfuscated interfaces, which cost extra security layers [43], otherwise they are vulnerable to all possible attacks (Criterion 1).

### 5.4. Newer PUF Architectures

There are other PUF architectures that have recently been introduced as an alternative solution for IoT devices. Transient Effect Ring Oscillator (TERO) PUFs [58,59] (see Figure 8) are an alternative to RO PUFs. Initial results [59,60] indicate that this architecture is not susceptible against cloning by electromagnetic analysis [61] or electromagnetic injections [62]. A TERO PUF structure is similar to that of an RO PUF, but is constructed from TERO cells that have two states: a transient oscillating state and a stable state [60]. The basic structure of a TERO PUF is an RS flip flop. By setting the init signal to one, the circuit starts oscillation for a short period of time.

Hybrid ring oscillator arbiter PUFs [63] and Public PUFs (PPUFs) [64] are other PUF architectures that have been proposed recently for use in the IoT. Hybrid ring oscillator arbiter PUFs combine RO PUFs with arbiter PUFs. They are reported to have very low power consumption [63], making them a good candidate for battery-operated IoT devices. Public PUFs (PPUFs) are another evolution of arbiter PUFs that use XORs instead of multiplexers. PPUFs can be used to implement public key protocols in IoT and have been reported to be resilient against side channel attacks, have good area efficiency, and consume little power [64].

There are also other PUF designs and solutions [65,66,67,68,69] that we do not address in this paper, although they look promising and have more stability and robustness in contrast to their predecessor. The main drawback of all these proposed PUF architectures is that they are relatively new, and their strengths and weaknesses are not yet fully understood. These new PUF architectures, according to the criteria presented earlier in this section, need to clarify whether they are good candidates for FPGA-based IoT devices or if they fit better for ASIC solutions. In addition, these next steps are recommended:Combination of these new PUFs with an authentication protocols to fulfil required IoT authentication scenarios (see Section 7).Integration of the solution in FPGAs with few computational resources, e.g., the Artix 7 or the Spartan family from Xilinx.Evaluation of the term “lightweight solution for the authentication process” after a clear report regarding the required computational resources.Collaboration with third-parties to reevaluate the achieved results.

Therefore, further investigations are required before using these new PUF architectures in real IoT systems.

## 6. PUF Protocols for IoT

Different authentication protocols that use a PUF have been proposed. In the following, we give an overview of existing protocols, taking into account different features that they support. We mainly focus on protocols that have been proposed explicitly for the IoT, but also take into account protocols that provide a feature that we find important for the IoT and that may be applied to it. A short summary is given in Figure 9.

Protocols are often not independent of the type of PUF (weak or strong) that they use. Protocols based on weak PUFs [16,40,70] typically require cryptographic methods (e.g., hashing, encryption, etc.) to compensate for CRP scarcity. This mostly negates the advantage of using a PUF in IoT. Therefore, we focus on protocols that **use strong PUFs** in this paper.

The first authentication protocol that used a (strong) PUF [15] (as described in Section 2) was based on the original assumption of a perfect PUF that can never be cloned or modeled [70]. As further research showed (see Section 3) this assumption proved to be incorrect. Especially when faced with machine learning analysis of CRPs [22], PUFs showed how fragile they can be. The most important task of a PUF authentication protocol therefore is to **be robust against machine learning attacks**.

Another important feature of PUF protocols is the ability to **cope with (partially) unstable PUFs**. Unlike initial PUF definitions suggest, responses are often not 100% stable, and protocols generally need error correction methods to cope with this [71].

Finally, it is especially important for IoT systems to **support mutual authentication**. Both communication partners should be authenticated against each other. Otherwise, IoT sensors might, e.g., send personal data about users to untrustworthy (unauthenticated) servers, and servers might accept faked sensor measurements from (unauthenticated) attackers. Since IoT devices often communicate directly, ideally, mutual authentication should be provided, not just between a device and a server, but also between two devices.

### 6.1. Early Protocols

In 2015, Delvaux et al. published a survey about authentication protocols that use strong PUFs [37]. Our study is based on this survey. In the following, we briefly present their findings, focusing on robustness against machine learning. Once machine learning attacks became known, a number of authors developed approaches to immunize against them [42,72,73,74,75]. Although their exact details differ, all these protocols follow the same process: after the device receives a challenge and generates a response, the protocols apply cryptographic primitives like hashing or encryption algorithms to the response and send it back to the server. Some protocols use Nonvolatile Memory (NVM) in this process [41,73,76,77,78,79,80,81,82]. However, implementing even a simple cryptographic algorithm costs hardware overhead, and NVM contradicts one of the original motivations for using PUFs, namely to not store any secret on the device. This is especially problematic for IoT systems, since, as discussed before, they have very little resources and are vulnerable to physical attacks. None of the early protocols were designed specifically for use in the IoT.

Since the original survey, a number of newer protocols have been proposed, addressing some of the open issues of the older ones. We discuss the most promising of these protocols in the following.

### 6.2. Mutual Authentication Protocol

Aman et al. [25] proposed a PUF-based authentication protocol that provides mutual authentication between devices in the IoT. The protocol covers two authentication scenarios: device-server mutual authentication and device-device mutual authentication. The device-server authentication scenario is as follows: each IoT device has an ID. First, the device sends its ID (let us assume $ID_{A}$) plus a random nonce $N_{1}$ to the server. The server selects a CRP ($C_{i}$,$R_{i}$) and generates a random number $R_{s}$. Then, the server generates an encrypted message $M_{A}$ = ($ID_{A}$,$N_{1}$,$R_{s}$${)}_{R_{i}}$. It then sends this message $M_{A}$ together with the challenge $C_{i}$ and a message authentication code $MAC_{A}$ = ($M_{A}$‖$R_{i}$‖$R_{s}$) to the IoT device. The IoT device applies $C_{i}$ to its local PUF to regenerate $R_{i}$, decrypts the message $M_{A}$ with it, and verifies the authenticity, integrity, and freshness of the message with the provided MAC. In the next step, the IoT device generates a new CRP ($C_{i+1}$,$R_{i+1}$) by using the new challenge ($C_{i+1}$ = H($R_{s}$‖$N_{A}$) with *H* being the Hamming distance function and $N_{A}$ another generated random number. Again, the device generates an encrypted message $M_{s}$ = ($ID_{A}$,$N_{A}$,$R_{i+1}$${)}_{R_{i}}$. $M_{s}$ and sends it with its $MAC_{s}$ to the server. The server verifies the message freshness with the MAC, regenerates $C_{i+1}$, and checks the correctness of the provided $R_{i+1}$. The device-device mutual authentication protocol is similar to this approach and uses the server to authenticate the two devices to each other. Since the resulting process is nearly the same as for device-server authentication, we do not discuss it again.

The main advantage of this protocol is the mutual authentication feature and the possibility for device-device authentication, which is very useful for many IoT applications. Its main shortcoming is that the authors still assumed a perfect PUF, and was is not clear if it was robust against machine learning attacks. Partially unstable PUFs were not considered. In addition, like earlier approaches, the protocol used cryptographic primitives, violating the original motivation for a PUF. Besides that, the hardware implementation overhead has not been reported.

### 6.3. Obfuscated Challenge Response Protocol

In contrast to the last approach, the obfuscated challenge response protocol [83] does not require encryption or other cryptographic primitives and is robust against machine learning attacks. However, it does not provide mutual authentication and is restricted to IoT devices authenticating against a server. Partially unstable PUFs are again not taken into account.

The main idea of the protocol is to obfuscate the direct relationship between challenges and responses by transferring only part of the challenge to a so-called Obfuscated (OB) PUF. As shown in Figure 10, an OB-PUF consists of a random number generator, a challenge control block, and a standard PUF, in this example an arbiter PUF.

When a partial challenge $C_{OB}$ is applied to the OB-PUF, the challenge control block asks the generator for a random number and combines it with $C_{OB}$ to create a full challenge *C*. This challenge is then applied to the PUF, and the response is given back. The full authentication process works as follows:First, in the enrollment phase, the server stores CRPs for partial challenges combined with all possible random numbers. Thus, for each partial challenge, it stores *n* possible responses.For authentication, the server sends a partial challenge $C_{OB1}$ to the device. The device applies it to its OB-PUF and sends back the generated response *R*.When the server receives *R*, it compares it with all possible responses $R_{em1}\ldots R_{emK}$ which have $C_{OB1}$ as part of their full challenge. If $R_{emN}$=R, N=1…K authentication is successful, otherwise authentication is rejected.

Because an attacker does not see full CRPs and never knows which random number was added to a partial challenge to generate a given response, this scheme makes it much more difficult for a machine learning attack to create a working model of the PUF.

### 6.4. Lockdown Protocol

Similar to the previous protocol, the lockdown protocol [84] focuses on providing resiliency against machine learning attacks without needing cryptographic primitives. It provides mutual authentication, however just between a device and a server, not between devices. In contrast to earlier approaches, lockdown can handle unstable PUFs. The authors proposed variants for strong and weak PUFs, but we restrict ourselves to the strong PUF version.

The protocol uses a strong XOR arbiter PUF, which by itself has been shown to be one of the PUF architectures most resilient against machine learning attacks (as discussed in Section 5.1). Again, the protocol has two phases (see Figure 11). In the enrollment phase, the server applies all possible challenges to the PUF by putting the PUF in a special XOR bypass mode. However, instead of storing the responses directly as in other approaches, the server uses machine learning to generate an authentication verification model $\hat{SPUF_{i}}$, which simulates the PUF. This $\hat{SPUF_{i}}$ model will be used for further authentication processes. At the end of the enrollment phase, the PUF is locked via an irreversible fuse or tamper-proof storage to ensure that bypassing XORs are no longer possible.

In the authentication phase, the device first sends its ID and a challenge $C_{D}$ to the server. The server selects a second challenge $C_{S}$ and applies $C_{S}\|C_{D}$ to a pseudo random number generator, which generates a new challenge <C>. $C_{S}$ essentially acts as a counter and is increased each time an authentication is done. This makes it impossible for attackers to gain enough data to train their models effectively. The server then applies <C> to $\hat{SPUF_{i}}$ and receives a response *r*, which it breaks up into two parts $r_{1}$ and $r_{2}$, such that $r=r_{1}\|r_{2}$. Then, the server sends $C_{S}$ and $r_{1}$ to the device. Using the received $C_{S}$, the device reconstructs <C> and applies it to its PUF. The result is $\tilde{r}={\tilde{r}}_{1}\|{\tilde{r}}_{2}$. Then, the device computes the fractional Hamming distance between $r_{1}$ and ${\tilde{r}}_{1}$. If it is beyond a given threshold, then authentication is aborted. Otherwise, the device sends ${\tilde{r}}_{2}$ to the server. Finally, the server computes the fractional Hamming distance between $r_{2}$ and ${\tilde{r}}_{2}$. Again, if they differ too much, the server aborts authentication. Otherwise, authentication terminates successfully.

## 7. Where Are We with PUFs in IoT?

Classical PUF architectures like RO PUFs promised to provide unclonable fingerprints, but were vulnerable to machine learning attacks and manipulated properties of their physical environment. Newer architectures have made clear progress towards solving these issues. Still, they are not understood in enough detail and need to be tested from different perspectives to evaluate their advantages and disadvantages. Regarding PUF protocols, the promise of PUFs was to provide lightweight secure authentication for IoT devices without the need for cryptography or secure memory. However, most of the proposed protocols use at least one of these two techniques. Recent approaches like [83,84] look promising. Still, there are open issues that will be discussed in the following.

Authentication is usually handled between a server and an IoT device. Some approaches offer mutual authentication, and most only support the devices authenticating against the server. In the IoT, authentication between devices and even between users and devices is important, as well. Figure 12 shows all possible authentication processes in IoT. If and how PUF can help here is mostly an open question. Authenticating without a server will probably be very difficult or even impossible. Note that the only protocol that supports device to device authentication (the mutual authentication protocol) does so using a server as a mutually trusted mediator. Not all these different authentication processes will be needed in all IoT scenarios, and some of them can be neglected or bridged (e.g., user-server through IoT device).

Regarding PUF architectures, an XOR arbiter PUF is robust, but not suitable for (dynamic) FPGAs and fits better in (static) ASICs. This feature can be considered a drawback for XOR arbiter PUFs because once deployed, they cannot be updated easily if new attacks are developed, which is quite probable in the IoT ecosystem. Ring oscillator PUFs are better suited for FPGAs, but they depend on physical context and must be made more robust against changes in the physical environment before they can be used reliably.

Machine learning attacks are an ongoing issue. Some approaches claim to be robust against them, but it is not clear enough how secure they really are. Current secure protocols are based on arbiter PUF. More work is needed to see if they also work with other PUF architectures like, e.g., RO PUFs.

In situations in which we just need pure authentication (e.g., smart locks), PUFs might be able to remove the necessity for encryption. However, in many scenarios, we need to communicate securely (e.g., to perform IoT device firmware updates), and then, we need encryption anyway. Here, PUFs may be interesting for key exchange or may act as good random number generators. Therefore, utilizing PUFs without encryption might be an interesting topic in research; however, in real use case scenarios, their applications are limited.

As discussed before, to counter replay attacks, protocols should never reuse CRPs and strong PUFs that support many CRPs should be used. However, due to the very long lifetime of IoT devices, it is quite challenging to estimate how many will be needed. At the same time, IoT devices must be inexpensive and use small PUFs, restricting the number of available CRPs. It is not clear what should happen if we run out of CRPs while the device is still used. Techniques to increase the number of CRPs for small PUFs like [7] are able to mitigate this challenge, but will not solve them completely.

Device management and setup comprise another challenge. A server needs to read out and store a large number of CRPs for each device before delivering this to a consumer. This can be a time-consuming process, and the resulting data (which can be roughly approximated as one terabyte per one million PUFs) must be stored securely and persistently. If they are lost, authentication is no longer possible. If the data are hacked, then the system immediately becomes insecure. This is another reason why we should be able to change the PUF even after the device is deployed, and this makes FPGAs a more attractive platform to implement PUFs.

Finally, in the case of the adversary gaining physical access to the device, PUFs may be in trouble. Although there is no secret that the attacker could read out, if the attacker holds on to the PUF, he/she can use it to answer challenges. If the attacker only has temporary access to the PUF, restricting the ability of the attacker to read out CRPS as quickly as possible can help. Still, this usually requires cryptographic primitives such as secure hashes or slows down valid authentications [13,37].

## 8. Conclusions

In this paper, we investigated PUF-based authentication solutions in IoT. Our focus was on IoT devices that have energy and computational resource concerns. We analyzed possible threats and how current PUF architectures, as well as PUF-based protocols address them. At this time, some companies have started to use PUFs in IoT (e.g., [50]), but overall, the field remains an active research area with many open challenges. PUF architectures need to become more mature. PUFs on ASICs have gained some publicity and show good characteristics. However, they cannot react to new attacks once they are deployed, and designing a tailor-made ASIC chipset for a secure PUF is costly. FPGA-based PUFs may provide a solution for this, but need to provide more resiliency against possible threats. The latest PUF-based protocols and designs are promising and overcome many of the shortcomings of their predecessors. To this day, there is however no protocol that fully addresses all challenges posed by the IoT, specifically with respect to the different types of authentication that are required and to the long lifetime of these systems. Regarding PUF design, recently, new solutions have been proposed [65,66,67,68,69]. These new solutions are progressing toward an evolutional path to address IoT security requirements as well.

Overall, we believe that it is important for researchers in the IoT field to take a step back, revisit the initial objectives behind using PUF in IoT, and re-evaluate the original use cases. Instead of replacing cryptography by PUF, PUFs might help in implementing it more efficiently, e.g., by creating encryption keys, better random numbers, or electronic signatures. This has been the focus of newer work done in the PUF community. It is recommended for the IoT community as well.

## Figures and Tables

**Figure 1 sensors-19-03208-f001:**
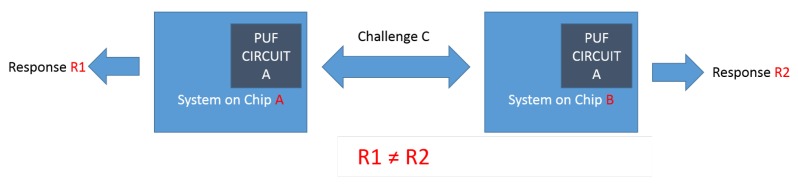
Two identical physical unclonable functions (PUFs) circuits on two different chips generate different responses.

**Figure 2 sensors-19-03208-f002:**
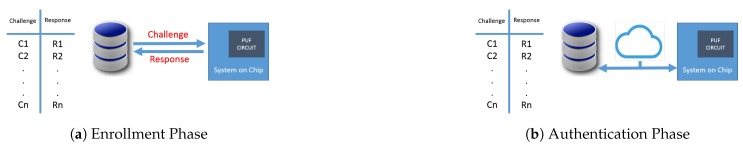
PUF authentication modes.

**Figure 3 sensors-19-03208-f003:**
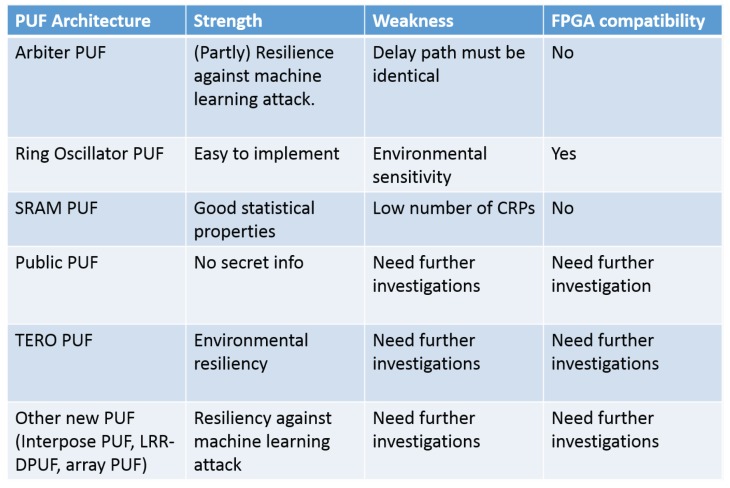
PUF architecture summary table. CRP, Challenge Response Pair; TERO, Transient Effect Ring Oscillator; LRR-DPUF, learning resilient and reliable digital PUF.

**Figure 4 sensors-19-03208-f004:**
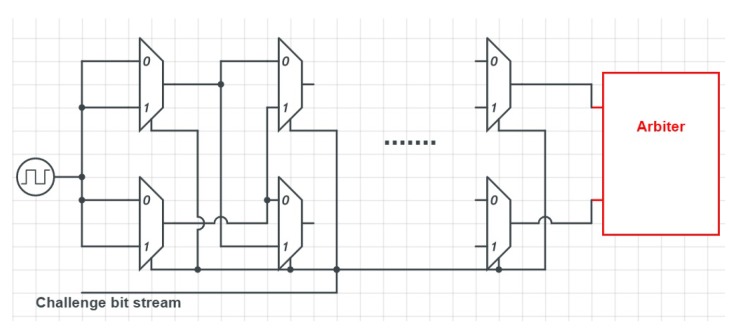
Arbiter PUF architecture. Adapted from [38].

**Figure 5 sensors-19-03208-f005:**
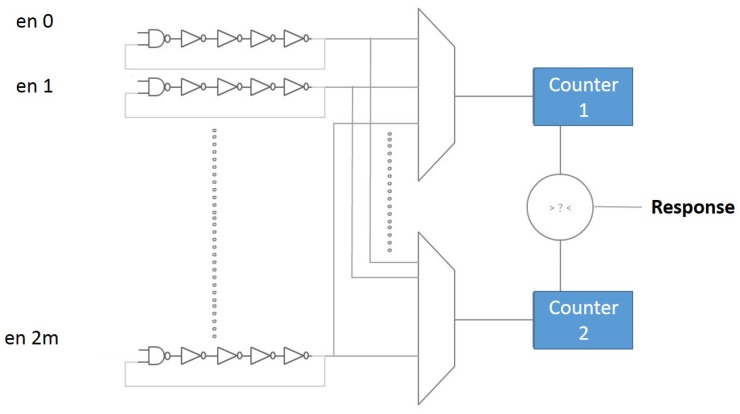
Ring oscillator architecture. Adapted from [7].

**Figure 6 sensors-19-03208-f006:**
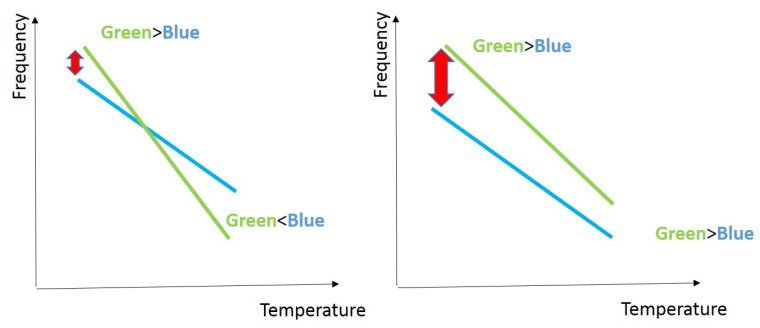
Bit flip due to temperature variation in unstable pairs. Adapted from [45].

**Figure 7 sensors-19-03208-f007:**
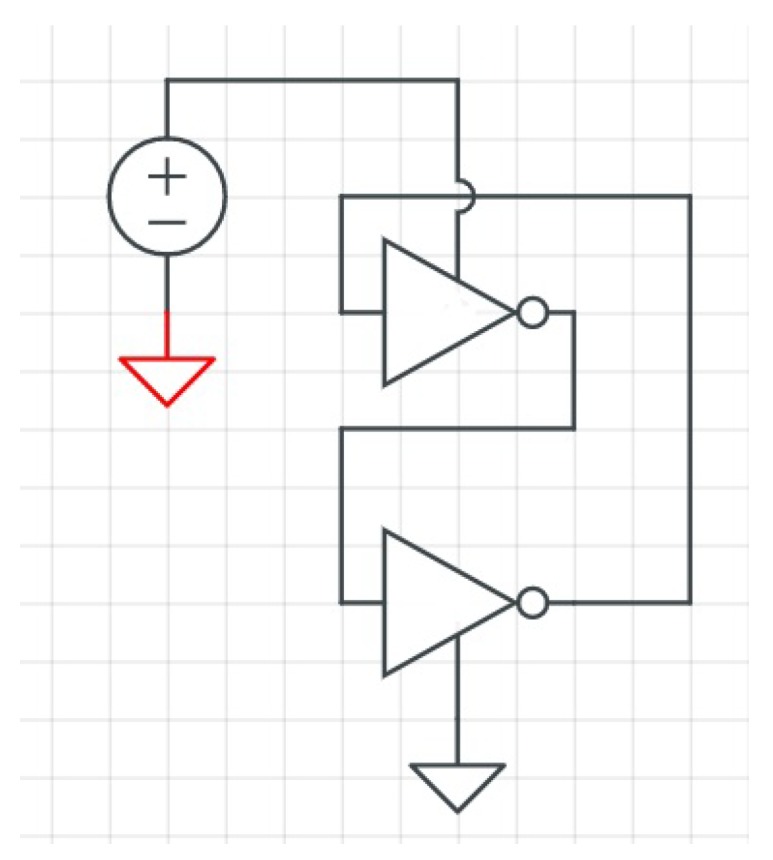
SRAM PUF architecture.

**Figure 8 sensors-19-03208-f008:**
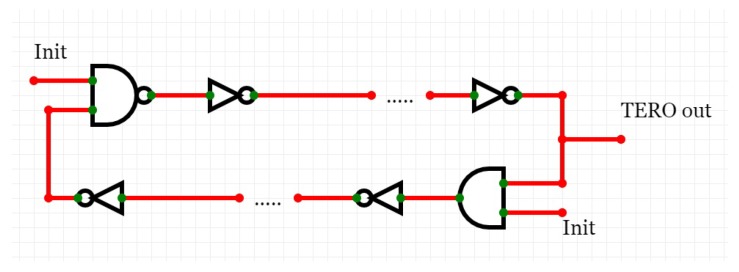
TERO PUF architectures. Adapted from [60].

**Figure 9 sensors-19-03208-f009:**
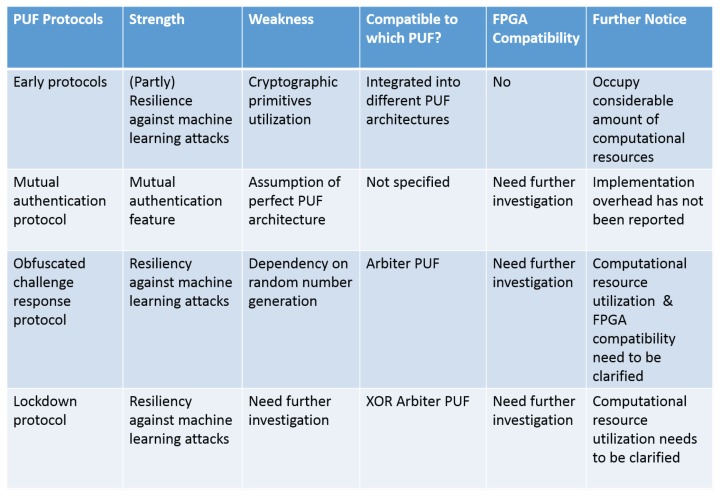
PUF protocols.

**Figure 10 sensors-19-03208-f010:**
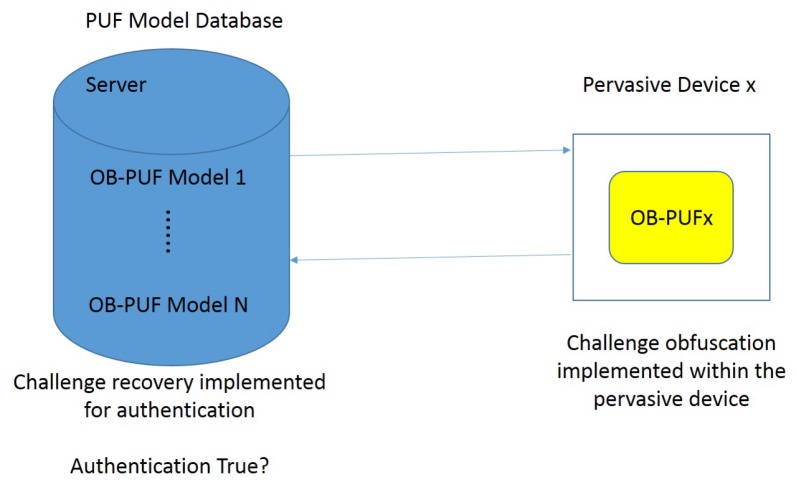
Obfuscated PUF structure. Adapted from [83].

**Figure 11 sensors-19-03208-f011:**
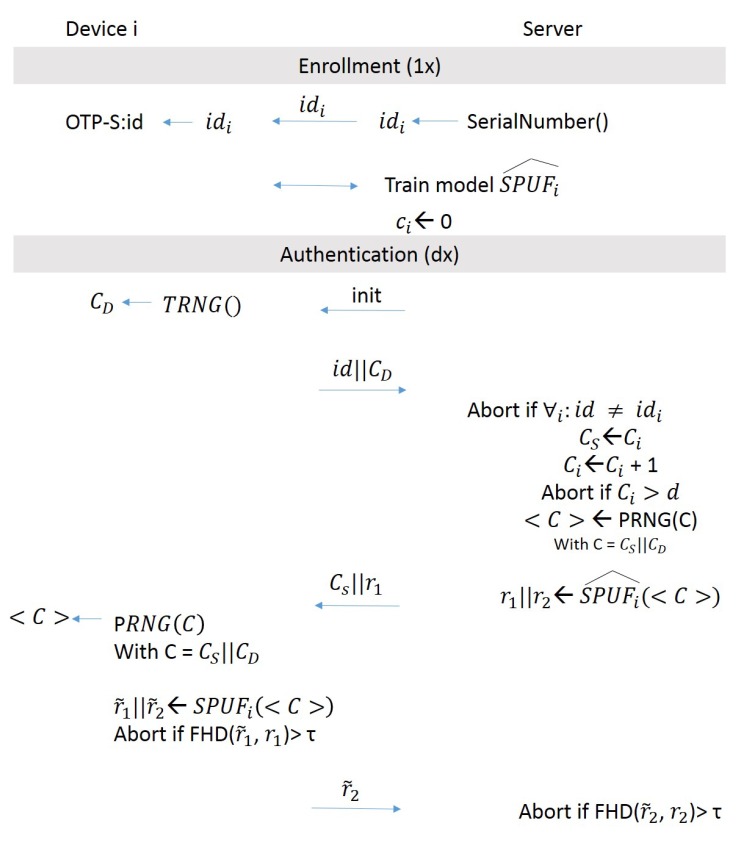
Lockdown protocol. Adapted from [84].

**Figure 12 sensors-19-03208-f012:**
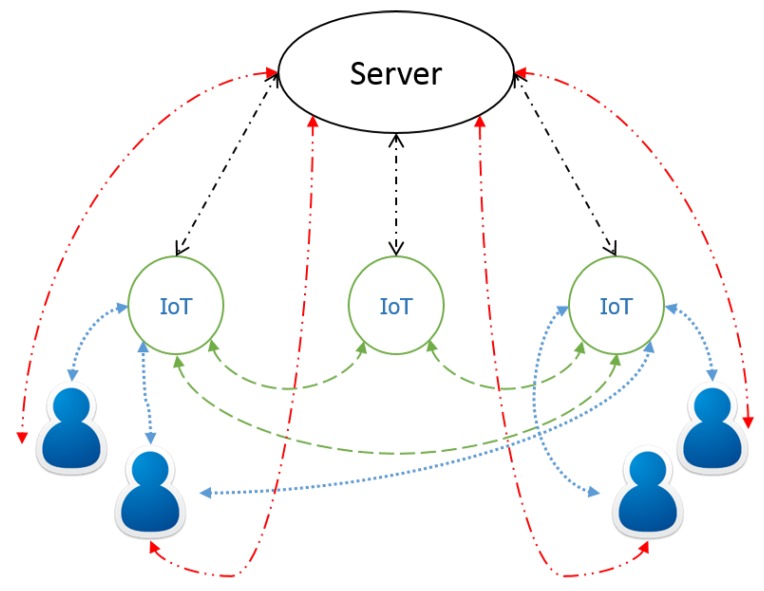
Possible authentications for IoT devices.

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
