# Peer review of "Physical Unclonable Functions in the Internet of Things: State of the Art and Open Challenges"

_sensors, 2019, doi:10.3390/s19143208_

Round 1

Reviewer 1 Report

Summary: This is a survey of PUF research. 

Strengths:

This paper specifically targets IoT application, which narrows down the area of to-be-studied research in the large PUF area. This makes it easy to digest.

Weaknesses:

In the introduction, it will be very helpful if the authors can summarize their contributions in this paper more concretely. This will help the reviewers understand the paper fast and also provide a chance for the authors to think about the paper contents thoroughly. 

Since the authors did not claim any contribution, I reviewed this paper as a survey. As a survey paper, one would like to see a very good summary of everything. A table or a figure that summarizes the comparison of attacks and corresponding defenses will be very appreciated. 

There are so many PUF structures and protocols. The authors' survey is clearly not complete. For example, a PUF whose security can be reduced to a well known computational hardness was proposed and implemented on FPGAs recently (See [1] and [2] below). A PUF that has been tested against all known attacks was introduced recently too (see [3]). There are much more PUF designs out there that the authors did not cover.  

Also I feel that the authors spent too many texts to introduce individual protocols. A survey paper just needs to briefly introduce each individual material.

Author Response

Open Review

English language and style

( ) Extensive editing of English language and style required 
(x) Moderate English changes required 
( ) English language and style are fine/minor spell check required 
( ) I don't feel qualified to judge about the English language and style 

Yes

Can be improved

Must be improved

Not applicable

Does the introduction provide sufficient background and include all relevant references?

( )

( )

(x)

( )

Is the research design appropriate?

( )

( )

( )

(x)

Are the methods adequately described?

( )

(x)

( )

( )

Are the results clearly presented?

( )

( )

(x)

( )

Are the conclusions supported by the results?

( )

(x)

( )

( )

Comments and Suggestions for Authors

Summary: This is a survey of PUF research. 

Strengths:

This paper specifically targets IoT application, which narrows down the area of to-be-studied research in the large PUF area. This makes it easy to digest.

Weaknesses:

In the introduction, it will be very helpful if the authors can summarize their contributions in this paper more concretely. This will help the reviewers understand the paper fast and also provide a chance for the authors to think about the paper contents thoroughly. 

The reviewer comments provided a comprehensive outlook regarding the required modifications. The first important issue was regarding our contribution in this paper we clarified and addressed it explicitly in the introduction (Section 1, new text is given in red).

Since the authors did not claim any contribution, I reviewed this paper as a survey. As a survey paper, one would like to see a very good summary of everything. A table or a figure that summarizes the comparison of attacks and corresponding defenses will be very appreciated. 

We agree, this is a very good idea. We added two tables to summarize our work, please see Section 5 and Section 6,new text is given in red

There are so many PUF structures and protocols. The authors' survey is clearly not complete. For example, a PUF whose security can be reduced to a well known computational hardness was proposed and implemented on FPGAs recently (See [1] and [2] below). A PUF that has been tested against all known attacks was introduced recently too (see [3]). There are much more PUF designs out there that the authors did not cover.  

Our goal in this paper was to provide the IoT community with an overview regarding PUFs, their application in IoT and the challenges that IoT developers should know about before integrating a PUF in their system. This was the main reason behind  selecting current journal(Sensor MDPI) for our paper. Of course there are much more resources regarding different varieties of PUFs and protocols. We tried to address those approaches which have been used/discussed by more than one source, and reached to appropriate level of maturity. We modified the section about PUF architectures and addressed new technologies in Section 5.4 and Section 7 (new text is given in red).

Also I feel that the authors spent too many texts to introduce individual protocols. A survey paper just needs to briefly introduce each individual material.

Since the target audience for this paper target is the IoT community and they might not be familiar with PUF protocols, we tried to describe  them in detail. furthermore, we received another reviewer comment who proposed to have more detailed descriptions about the protocols. Therefore we decided not to change this part of the paper.

Submission Date

29 April 2019

Date of this review

06 May 2019 23:47:55

Reviewer 2 Report

Overall summary:

In the paper, the authors focus on introducing PUFs and analyze their challenges and current solutions for IoT applications. Important materials, including the threats, defenses, PUF architectures, and authentication protocols, are covered by the paper. Overall, the paper has a good readability and covers critical materials in the area.

Weaknesses and suggestions:

1) In line 125 - 127, the authors make a statement that encrypting the CRPs are especially suitable if the number of CRPs is very limited, e.g., for weak PUFs. However, even for strong PUFs, encrypting CRPs can be helpful, right.

2) The architectures of the strong PUFs introduced in the paper only cover the arbiter PUF and the RO PUF as well as their modifications. In fact, there are other strong PUF designs, including the Digital PUF [1] and SCA PUF [2], proposed in the literature that are resilient to machine learning attacks. It would be helpful if the authors can cover these recent papers.

3) The review on the PUF protocols are important. It would be very helpful if the authors can use figures similar to Figure 11 to illustrate the mutual authentication protocol and the obfuscated challenge response protocol.

4) There are some typos in the paper. For example, in line 92, it should be "still require special equipments". The authors need to do a spell check and a grammar check. Also, in line 179, "fa" and "f2m" are not explained.

[1] Miao, Jin, et al. "LRR-DPUF: Learning resilient and reliable digital physical unclonable function." 2016 IEEE/ACM International Conference on Computer-Aided Design (ICCAD). IEEE, 2016.

[2] Xi, Xiaodan, et al. "Strong subthreshold current array PUF with 2 65 challenge-response pairs resilient to machine learning attacks in 130nm CMOS." 2017 Symposium on VLSI Circuits. IEEE, 2017.

Author Response

Open Review 2

(x) I would not like to sign my review report

( ) I would like to sign my review report

English language and style

( ) Extensive editing of English language and style required

( ) Moderate English changes required

(x) English language and style are fine/minor spell check required

( ) I don't feel qualified to judge about the English language and style

Yes

Can be improved

Must be improved

Not applicable

Does the introduction provide sufficient background and include all relevant references?

(x)

( )

( )

( )

Is the research design appropriate?

(x)

( )

( )

( )

Are the methods adequately described?

(x)

( )

( )

( )

Are the results clearly presented?

( )

(x)

( )

( )

Are the conclusions supported by the results?

( )

(x)

( )

( )

Comments and Suggestions for Authors

Overall summary:

In the paper, the authors focus on introducing PUFs and analyze their challenges and current solutions for IoT applications. Important materials, including the threats, defenses, PUF architectures, and authentication protocols, are covered by the paper. Overall, the paper has a good readability and covers critical materials in the area.

Thank you very much for your review. We did our best to address your comments to make the paper stronger.

Weaknesses and suggestions:

1) In line 125 - 127, the authors make a statement that encrypting the CRPs are especially suitable if the number of CRPs is very limited, e.g., for weak PUFs. However, even for strong PUFs, encrypting CRPs can be helpful, right.

We modified the text according to the reviewer comment. Please see Section 4, new text is given in red.

2) The architectures of the strong PUFs introduced in the paper only cover the arbiter PUF and the RO PUF as well as their modifications. In fact, there are other strong PUF designs, including the Digital PUF [1] and SCA PUF [2], proposed in the literature that are resilient to machine learning attacks. It would be helpful if the authors can cover these recent papers.

Our goal in this paper was to provide the IoT community with an overview regarding PUFs, their application in IoT and the challenges that IoT developers should know about before integrating a PUF in their system. This was the main reason behind  selecting current journal(Sensor MDPI) for our paper. Of course there are much more resources regarding different varieties of PUFs and protocols. We tried to address those approaches which have been used/discussed by more than one source, and reached to appropriate level of maturity. We modified the section about PUF architectures and addressed new technologies in Section 5.4 and Section 7 (new text is given in red). According to the reviewer comment, we clarified our systematic approach and described it in more detail (please see Section 1, new text is given in red).

3) The review on the PUF protocols are important. It would be very helpful if the authors can use figures similar to Figure 11 to illustrate the mutual authentication protocol and the obfuscated challenge response protocol.

Unfortunately, we received other review feedback to reduce the descriptions and contents about PUF protocols, therefore, we decided to not change this section.

4) There are some typos in the paper. For example, in line 92, it should be "still require special equipments". The authors need to do a spell check and a grammar check.

Thank you for the suggestion. We did another check and asked a native speaker to give us feedback. We hope that we caught everything major.

Also, in line 179, "fa" and "f2m" are not explained

We modified Section 5.2 (new text is in red) to explain these parameters.

Submission Date

29 April 2019

Date of this review

20 May 2019 08:22:03

Reviewer 3 Report

At best, the paper provides an interesting set of references. The comparisons are not really comprehensive, deep or systematic and the paper does not provide significant new insights. As an overview paper it is not expected to provide new technical results, but should generate new insights e.g. by rigorous comparisons. Even if I acknowledge the effort, I cannot recommend to accept the paper.

A subset of critical statements are for example:

"Recently, Physical Unclonable Functions (PUFs) have been proposed as a lightweight, cost efficient and ubiquitous solution for device authentication."
The idea of PUFs is not new. The first papers are over 15 years old.

The proposition "Good physical protection and anomaly detection, e.g. by detecting unusual device mobility, can solve the issue. Since physical protection is a well established industrial process, we propose to use it for all IoT devices that need to be secure." is quite challenging. It would be interesting to see a suggestion how to address this issue for low-cost IoT devices.

The ideas of PUFs not responding to all challenges and of having slow response patterns is not new.

"In addition, the architecture of the used PUF can provide robustness against non-invasive side channel attacks" how can this be achieved?

5.2.: using non-exclusive RO comparisons as strong PUFs is a bad idea. Yu and Devadas (GOMACtech 2010) proposed a solution that slightly improves the issue.

7. "Instead of replacing cryptography, PUFs might help in implementing it more efficiently, e.g. by creating encryption keys, better random numbers, or electronic signatures." This is exactly what the PUF community looked into in detail over quite a while.

Author Response

Open Review 3

English language and style

( ) Extensive editing of English language and style required 
(x) Moderate English changes required 
( ) English language and style are fine/minor spell check required 
( ) I don't feel qualified to judge about the English language and style 

Yes

Can be improved

Must be improved

Not applicable

Does the introduction provide sufficient background and include all relevant references?

(x)

( )

( )

( )

Is the research design appropriate?

( )

( )

(x)

( )

Are the methods adequately described?

( )

( )

(x)

( )

Are the results clearly presented?

( )

( )

(x)

( )

Are the conclusions supported by the results?

( )

( )

(x)

( )

Comments and Suggestions for Authors

At best, the paper provides an interesting set of references. The comparisons are not really comprehensive, deep or systematic and the paper does not provide significant new insights. As an overview paper it is not expected to provide new technical results, but should generate new insights e.g. by rigorous comparisons. Even if I acknowledge the effort, I cannot recommend to accept the paper.

Thank you for your review. It helped us realise that we did not make the goal of our paper clear enough and that the paper needs to be improved. Our focus is not to provide a full survey of PUFs for a security audience but to provide members of the IoT community with an overview regarding PUF, its promises and challenges they shall expect when integrating a PUF in their system. This was the main reason behind selecting the current journal (Sensor MDPI) for our paper. Of course there are much more resources regarding different varieties of PUF and protocols. However, to be useful for an application-centric IoT audience, we tried to focus on approaches/techniques that have reached an appropriate level of maturity and are used by multiple groups. We realise that we did not state this explicitly in our paper and according to the reviewer comment, we improved it and clarified our systematic approach (see Section 1 and 7, new text is given in red). Please note, that we also asked a native speaker to give us feedback on grammar and typos. We hope that we caught everything major and that we were able to address all your comments successfully.

A subset of critical statements are for example:

"Recently, Physical Unclonable Functions (PUFs) have been proposed as a lightweight, cost efficient and ubiquitous solution for device authentication."
The idea of PUFs is not new. The first papers are over 15 years old.

The sentence was referring to IoT devices, we modified it to make this clearer. Please see the abstract and Section 1, new text is given in red)

The proposition "Good physical protection and anomaly detection, e.g. by detecting unusual device mobility, can solve the issue. Since physical protection is a well established industrial process, we propose to use it for all IoT devices that need to be secure." is quite challenging. It would be interesting to see a suggestion how to address this issue for low-cost IoT devices.

In this section we referred to established techniques such as glueing and/or adding movement detection, e.g. with a gyroscope. We added a short description regarding this issue to clarify it more. Of course both techniques only mitigate the risk, but do not eliminate it. Please see Section 4, new text is given in red

The ideas of PUFs not responding to all challenges and of having slow response patterns is not new.

That is of course very true. We did not want to give the impression that these are new ideas. We added citations for solutions which already perform a similar process. Our statement was meant for IoT developers. We explicitly changed this phrase to clarify it in a better way. Please see Section 7,  new text is given in red

"In addition, the architecture of the used PUF can provide robustness against non-invasive side channel attacks" how can this be achieved?

Thanks for the comment. The most important non-invasive side channel attack for IoT devices is to modify environmental parameters to e.g. increase the PUF error rate and thus reduce the number of useable CRPs. Thus, higher robustness can be achieved by making the PUF architecture robust against such environmental variations. We modified this section accordingly (see Section 4, new text is given in red)

5.2.: using non-exclusive RO comparisons as strong PUFs is a bad idea. Yu and Devadas (GOMACtech 2010) proposed a solution that slightly improves the issue.

You are right of course. Our example was not choses very well. We changed it accordingly.

7. "Instead of replacing cryptography, PUFs might help in implementing it more efficiently, e.g. by creating encryption keys, better random numbers, or electronic signatures." This is exactly what the PUF community looked into in detail over quite a while.

Again, we agree with the reviewer and we realise now that our paper was not very clear. Our goal was to let the (non-security) IoT community know about this trend in the PUF community and to tell them that PUFs may still be useful for IoT systems, just not in the way many people still believe. We changed the sentence to make this clearer. Please see Section 8, new text is given in red.

Submission Date

29 April 2019

Date of this review

06 May 2019 15:27:26

Round 2

Reviewer 1 Report

The paper has been improved a lot after the revision. I appreciate the efforts taking by the authors. 

One remaining issue: 

The authors keeping talking about that many recently proposed PUFs need further investigation to find our their advantages and disadvantages. I think the papers which proposed these designs have mentioned their advantages and disadvantages, so the authors can restate the claims from these papers and clearly point out what are the remaining aspects that not mentioned/validated in their original papers. 

Author Response

round 2 Reviewer 1

Open Review

(x) I would not like to sign my review report

( ) I would like to sign my review report

English language and style

( ) Extensive editing of English language and style required

( ) Moderate English changes required

(x) English language and style are fine/minor spell check required

( ) I don't feel qualified to judge about the English language and style

Yes

Can be improved

Must be improved

Not applicable

Does the introduction provide sufficient background and include all relevant references?

(x)

( )

( )

( )

Is the research design appropriate?

( )

( )

( )

(x)

Are the methods adequately described?

( )

(x)

( )

( )

Are the results clearly presented?

( )

(x)

( )

( )

Are the conclusions supported by the results?

( )

(x)

( )

( )

Comments and Suggestions for Authors

The paper has been improved a lot after the revision. I appreciate the efforts taking by the authors.

One remaining issue:

The authors keeping talking about that many recently proposed PUFs need further investigation to find our their advantages and disadvantages. I think the papers which proposed these designs have mentioned their advantages and disadvantages, so the authors can restate the claims from these papers and clearly point out what are the remaining aspects that not mentioned/validated in their original papers.

Thanks for the comment. We are very happy that you agree with our changes. We really think that the changes helped the paper a lot - many, many thanks, once again!

Regarding your remaining issue: Point taken. We once again went through the new PUF architectures. According to the specified criteria which we stated in Section 5,  we added our recommendation regarding the required actions for these new PUFs. Please see Section 5, new added lines are in blue.

Submission Date

29 April 2019

Date of this review

06 Jun 2019 20:32:19

Reviewer 3 Report

The edits and explanations make the intention behind the paper more clear now. However, if it is intended for the IoT community, the paper is lacking a motivation and comparison to other technologies, why an IoT system designer should use a PUF.

Author Response

Second round-reviewer 3

(x) I would not like to sign my review report

( ) I would like to sign my review report

English language and style

( ) Extensive editing of English language and style required

( ) Moderate English changes required

(x) English language and style are fine/minor spell check required

( ) I don't feel qualified to judge about the English language and style

Yes

Can be improved

Must be improved

Not applicable

Does the introduction provide sufficient background and include all relevant references?

(x)

( )

( )

( )

Is the research design appropriate?

( )

( )

(x)

( )

Are the methods adequately described?

( )

( )

(x)

( )

Are the results clearly presented?

( )

( )

(x)

( )

Are the conclusions supported by the results?

( )

( )

(x)

( )

Comments and Suggestions for Authors

The edits and explanations make the intention behind the paper more clear now. However, if it is intended for the IoT community, the paper is lacking a motivation and comparison to other technologies, why an IoT system designer should use a PUF.

First of all, thanks for your feedback. We really appreciate the time and effort that you are investing in our paper! As you mentioned, the motivation was not discussed enough. We modified it in Section 1 and gave more detail, including citing how security can be addressed without a PUFG in IoT. Please see the modifications in Section 1 in blue text.

Submission Date

29 April 2019

Date of this review

06 Jun 2019 13:50:53

Round 3

Reviewer 1 Report

The authors have addressed my concerns. I support acceptance. 

Reviewer 3 Report

Adressing the IoT community properly would require more of a reframing of the paper than just changing one paragraph in the introduction. I still do not see the contribution of the paper for the reader that would justify the publication in a quality scientific journal.